# Microstructure, Mineralogical Characterization and the Metallurgical Process Reconstruction of the Zinc Calcine Relics from the Zinc Smelting Site (Qing Dynasty)

**DOI:** 10.3390/ma14082087

**Published:** 2021-04-20

**Authors:** Ya Xiao, Wenli Zhou, Linheng Mo, Jianli Chen, Meiying Li, Shaojun Liu

**Affiliations:** 1State Key Laboratory for Powder Metallurgy, Central South University, Changsha 410083, China; hnkgxy@163.com; 2Cultural Relics and Archaeology Institute of Hunan, Changsha 410083, China; laomo2009@sina.com (L.M.); limeiying402@gmail.com (M.L.); 3Hunan Key Laboratory for Scientific Archaeology and Conservation Science, Changsha 410083, China; 4Institute for the History of Natural Sciences, Chinese Academy of Sciences, Beijing 100190, China; juliazhouwl@gmail.com; 5School of Archaeology and Museology, Peking University, Beijing 100871, China; jianli_chen@pku.edu.cn

**Keywords:** archeological evidence, zinc metallurgy, zinc calcine, roasting, characterization, experimental reconstruction

## Abstract

The smelting of zinc is considered as one of the most challenging technologies in ancient civilization. Compared with non-sulfide zinc ores, the smelting of zinc sulfide ores is more complicated since they have to be roasted before smelting. The technological smelting process of the ancient zinc metallurgy technology has been studied and partly reconstructed. However, the roasting technology, including the roasting conditions and involved metallurgical processes, is still unclear. The discovery of the zinc smelting site of Doulingxia dates back to the Qing dynasty (CE 1636–1912), and for the first time provides us with critical archeological evidence to gain an insight into the roasting technology in ancient zinc metallurgy technology. In this paper, the microstructure and mineralogical features of the zinc calcine relics found at the Doulingxia site were characterized by X-ray diffraction (XRD) and scanning electron microscopy coupled with X-ray energy dispersive spectrometer (SEM-EDS). To reconstruct the metallurgical process, the original roasting temperature of the unearthed zinc calcine was estimated by thermogravimetric analysis and differential thermal analysis (TG-DTA), combined with reheating experiments and phase composition analysis as well as microstructural analysis. The simulation experiments were conducted to reconstruct the roasting process. The results indicated that the original roasting temperature of the unearthed zinc calcine should be in a range of 650–850 °C, most probably near 750 °C. As long as the retention time is long enough, all sphalerite can be oxidized when the roasting temperature is above 650 °C. The final roasting products mainly include tiny porous particles of ZnFe_2_O_4_, Fe_2_O_3_, PbSO_4_, and ZnO. These findings are helpful to reconstruct the ancient zinc metallurgy technology of zinc sulfide ores.

## 1. Introduction

The smelting of zinc is considered as one of the most challenging technologies in ancient civilization since the reduction temperature of zinc oxide ores is very close to the boiling point of metallic zinc [1,2,3]. Although zinc ores are widely distributed, often associated with other metals, such as lead and silver, zinc is one of the last major metals to be produced [4]. Before the establishment of modern zinc smelting technology in Europe, only India and China could produce metallic zinc on a large scale [1].

Ancient India is the earliest country recorded by literature in which metallic zinc was produced [1,5]. It has been revealed by archeological evidence that the metallic zinc was distilled and condensed by using specially designed retorts ~1000 years ago in Zawar, northwest India [5]. Primary sources for zinc smelting in Zawar mainly come from the sulfidic ore deposits of the Aravalli Hills, mainly constituted by sphalerite (ZnS), galena (PbS), and pyrite (FeS_2_) [6]. However, the smelting residues show that the metal was obtained from smelting of oxidized zinc ores [7,8]. The ore is likely to have been roasted at 900–950 °C to convert the starting materials to oxides [9]. However, neither remains of any kilns or ovens suitable for the oxidizing stage nor any roasting products were found in Zawar [2].

In contrast, the ancient Chinese zinc pyrometallurgical technology is mainly based on the smelting of non-sulfide zinc ores, which include smithsonite (ZnCO_3_) and hemimorphite [Zn_4_Si_2_O_7_(OH)_2_(H_2_O)] [4,10]. The archaeological evidence [11,12] and scientific analyses [13,14,15] showed that the ores distilled in Chongqing, southwest China, in the Ming (CE 1368–1644) and Qing (CE 1636–1912) dynasties, were mainly oxidized ores. Moreover, the investigations of traditional Chinese zinc smelting technology indicate that, until the late 20th century, zinc metallurgy in Western Central China, including the provinces of Yunnan [16], Guizhou [17], and Sichuan [18,19], was still limited to the non-sulfide ores.

In recent years, several zinc smelting sites have been found in Guiyang County, Hunan Province in central south China [20]. Among these sites, the most representative ones are the Tongmuling site and the Doulingxia site, dated back to the Qing dynasty. Significantly distinguished from the zinc smelting sites found in Chongqing [11,12,13,14,15], a large number of cylindrical hearths (Figure 1a) were found to be distributed on the edge of the smelting area in the sites [20]. Our previous study reveals that in these cylindrical hearths, a lengthy roasting process was conducted to oxidize the zinc sulfide ores at a lower temperature before distilling [21]. In addition, a basket of well-preserved reddish-brown powders confirmed as the products of the roasted zinc sulfide ores has been found at one of the house foundations in the Doulingxia site (Figure 1b). The distilling retorts and technological smelting process of zinc sulfide ores in Guiyang County have been reconstructed [21]. However, the roasting technology, including the roasting conditions and the metallurgical processes during roasting are still unclear.

It should be mentioned that the shape and arrangement of the roasting hearths found in the zinc smelting sites of Guiyang are similar to those used in the Songbai Traditional Zinc Smelting Plant in Hunan province. The plant started to carry out zinc smelting in 1905 by technicians recruited from Guiyang. The roasting method in Guiyang in the Qing dynasty period could likely be consistent with that in the Songbai zinc smelting plant. It was recorded that the entire roasting process in the Songbai Traditional Zinc Smelting Plant in Hunan province usually takes 21 days, requiring repeated calcination and crushing three times [21,22]. It can be seen that the roasting process is very time-consuming. In contrast, in present-day industrial zinc smelting, zinc sulfide ores are usually roasted in the roasting furnace at temperatures exceeding 900 °C for several hours [23,24]. For example, roasting in the fluid-bed roaster at approximately 950 °C for 6 h can convert up to 94% of sulfur contained in the ore to sulfur dioxide [25].

The zinc calcine remains found in the Doulingxia site, for the first time, provide us with critical archeological evidence to have an insight into the roasting technology of zinc sulfide ores. The present investigation aims to clarify the metallurgical process involving the roasting technology in pre-modern zinc production using sulfide zinc ores. The main issues discussed include the mineralogical and microstructural characterization of the unearthed calcine, the estimation of the roasting temperature, and the reconstruction of the chemical reactions that occur during the roasting process.

## 2. Materials and Methods

### 2.1. Materials

The unearthed zinc calcines from the Doulingxia site are reddish-brown powders that have been carefully roasted and crushed. The ores used for the reconstruction experiments come from the Huangshaping Lead-Zinc Mine, which is a large polymetallic deposit that was mined for an extended period in the Qing Dynasty [26,27], about 5 km away from the Doulingxia site. The ores used for the experiments were selected according to the analytical results of the unearthed calcines and the ores found in the Tongmuling site [21]. To avoid the inhomogeneity of the mineral content in the ores affecting the results of the experiments, and to shorten the retention time, the ores were crushed to particles with average diameters in a range from 2 mm to 1 cm.

### 2.2. Characterization of the Unearthed Zinc Calcine

This research is based on the thoroughly comprehensive characterization of the unearthed zinc calcine. The mineralogical and microstructural features of the unearthed calcine particles were firstly characterized by SEM-EDS, combined with XRD. To provide basic information for the later simulation experiments, the analysis focused on the mineral composition of the ores before roasting, the metallurgical reactions that occurred during roasting, and the microstructure and composition of the finally roasted products.

### 2.3. Estimate of the Roasting Temperatures

In this investigation, the method of reheating combined with XRD phase analysis was used to confirm the original roasting temperature of the calcine. Ignoring the affection of the weathering product during burial, when the unearthed calcine is reheated to a higher temperature than the original roasting temperature, the phase composition, and microstructure of the calcine will change significantly. The reheating temperatures were determined by the TG-DTA analysis results. The specific method was: insertion of 100 g of unearthed ore sand samples in a muffle furnace; reheating at the pre-set temperature for 3 h; mineralogical and textural analysis of the cooled samples by XRD and SEM-EDS.

### 2.4. Experimental Reconstruction of the Roasting Process

Experimental reconstruction was used in this research to decipher complex ancient roasting technologies. All experiments were conducted in a muffle furnace. Corundum crucibles 12 cm long and 6 cm wide were used as reaction containers. The air flow and redox conditions within the furnace could not be controlled, although presumably oxidizing conditions were maintained during the experimental runs as determined by the analytical results of the unearthed zinc calcine. The experimental temperatures were determined by the above preliminary estimation results, and the retention time was adjusted according to the actual roasting situation. The simulated roasting products were characterized by WDXRF, XRD, and SEM-EDS.

It should be noted that the entire roasting process in the Songbai Traditional Zinc Smelting Plant in Hunan province requires repeated calcination and crushing for three cycles, which takes 21 days in total [28]. The final state of the experimental products should be consistent with the unearthed zinc calcine. By comparing the chemical composition, phase composition, and microscopic morphology of the experimental roasting products and the unearthed zinc calcine, the original roasting temperature was further confirmed. Besides, the main metallurgical reactions that occurred during the roasting process were discussed.

The roasting degree is quantified by the degree of desulfurization of the simulation zinc calcine. The degree of desulfurization (D_s_) is given by the equation below:(1)$$D_{s}=S_{0}-S/S_{0}\times 100$$where S_0_ is the sulfur content of the zinc ores for reconstruction experiments before roasting and S is the sulfur content of the final experimental roasting products.

### 2.5. Characterization

The unearthed zinc calcines, the ores for the experiments, and the calcines produced by reconstruction experiments were characterized by the following analytical methods and techniques.

#### 2.5.1. Scanning Electron Microscopy Coupled with X-ray Energy Dispersive Spectrometer (SEM-EDS)

The microstructure and chemistry were characterized by the scanning electronic microscope (Zeiss EVO MA 10, Jena, Germany) and X-ray energy dispersive spectrometer (Brooke Xflash 6130, Berlin, Germany), respectively. The powdered samples of the unearthed zinc calcine and the simulated zinc calcine, were cold-inlaid with epoxy resin. Then, they were polished after solidification. The surface of the samples were sprayed with gold after cleaning and drying. The acceleration voltage applied to all analyses was 20 kV and the working distance was about 10.5 mm. The EDS results were assessed by semiquantitative analysis.

#### 2.5.2. Thermogravimetric Analysis and Differential Thermal Analysis (TG-DTA)

Thermal transformation of the unearthed zinc calcine was recorded with TG-DTA (SDT Q600, New Castle, TA, USA). The temperature range was 25–1010 °C with a heating rate of 10 °C/min in argon. The reference material was α-Al_2_O_3_.

#### 2.5.3. X-ray Diffraction (XRD) Analysis

The mineralogical components of the samples were determined by an X-ray diffractometer (Rigaku D/max 2500 PC, Tokyo, Japan) equipped with Cu K_α_ radiation and secondary graphite monochromator using dry and finely ground powders. The tube voltage and current were 40 kV and 40 mA, respectively. The scanning angle range (2θ) was in the range from 5 to 75° with a scanning speed of 8°/min.

#### 2.5.4. Wavelength Dispersive X-ray Fluorescence Measurement (WDXRF)

The chemical content of the samples was analyzed by a WDXRF spectrometer (Axios-Advanced, Malvern Analytical, Almelo, The Netherlands). The samples for WDXRF characterization were prepared from finely ground powders, which were further pressed into sheets by mixing a certain amount of boric acid.

## 3. Results and Discussion

### 3.1. Microstructure and Mineralogical Characterization of the Unearthed Zinc Calcine

The mineralogical composition of the unearthed zinc calcine includes major amounts of franklinite (ZnFe_2_O_4_), cerussite (PbCO_3_), and willemite (Zn_2_SiO_4_), as well as minor amounts of hemimorphite [Zn_4_Si_2_O_7_(OH)_2_(H_2_O)], anglesite (PbSO_4_), fluorite (CaF_2_), quartz (SiO_2_), and hematite (Fe_2_O_3_) [21]. Figure 2 shows the general morphology and microstructure of the unearthed zinc calcine. There are no-table differences in composition and morphology between different particles. The calcine particles can be divided into three categories according to their morphologies: compact angular-shaped particles, cracked particles, and porous particles.

The compact angular-shaped particles are homogeneous, mainly including fluorite (particle 1), quartz (particle 2), and aluminosilicate (particle 3) particles (Figure 2a). It can be inferred that these particles, which have excellent thermal stability, are directly derived from the ores and do not react during the roasting process. These dense, unreacted particles occur as tiny particles (~200 μm) (Figure 2b), indicating that the ore has been crushed.

The cracked particles are heterogeneous and partially roasted, since they still retain the angular-shaped morphology of the original particle. A partially reacted sphalerite particle occurs at the lower middle of Figure 2a (particle 4). The particle is slightly oxidized and divided into several parts by the cracks. The partially reacted phase contains a high amount of Fe, as well as minor amounts of Pb and Mn; the average analysis gives 51.2 wt.% Zn, 32.4 wt.% Fe, 4.0 wt.% S, 3.5 wt.% Pb, and 3.4 wt.% Mn. The surface and the portions along the cracks have been converted to a (Fe, Zn) oxide which contains small amounts of Zn (~3.0 wt.%). Another cracked particle is composed of Zn_2_SiO_4_, (Fe, Zn) O, Pb-Zn-As oxide, and Zn-Si-Al oxide phases (Figure 2a, particle 5).

The porous particles can be considered as the final products of the zinc calcine after complete reaction. As shown in Figure 2a, a large porous particle (~1 mm) (particle 6) mainly consists of tiny grains of ZnFe_2_O_4_, minor amounts of Zn_2_SiO_4_, as well as trace amounts of Fe-Zn-Pb oxide, PbSO_4_ or PbCO_3_ particles. The crudely spherical shapes, the porous texture and the heterogeneous composition imply that these large porous particles are formed by agglomeration of the tiny particles [29]. In addition, porous particles can also be found in the tiny zinc calcine particles. These tiny porous particles are mainly composed of Fe_2_O_3_, ZnFe_2_O_4_, Zn_2_SiO_4_, and PbSO_4_ or PbCO_3_. It is noted that these tiny particles still retain the angular shaped morphology of the original particles (Figure 2b).

Occasionally, partially reacted sphalerite particles are detected in the calcine. Figure 3 shows the morphology and the composition as well as the distribution of O, S, Zn, Fe, Pb, and Si in one of these particles. As shown in Figure 3a, the particle shows a core-shell morphology. The core of the particle is compact, while the shell is much more porous. As shown in Figure 3b, the core (zone A) of the particle is slightly reacted sphalerite, which contains 61.0 wt.% Zn, 31.7 wt.% Fe and 4.5 wt.% S. The brighter rim (zone B), which wrapped the core and divided the core into two parts, is Fe-Pb-Zn-Si-O phase and contains approximately 17.3 wt.% Pb with a trace amount of Cu (~1.6 wt.%). The adjacent dark grey rim (zone C) is a compact Fe-Zn-Si-O phase, with more Zn and less Pb than the brghter rim. The next grey rim (zone D) and the outer porous portion (zone E) separated from the core are a mixture of ZnFe_2_O_4_ and minor Zn_2_SiO_4_.

The distributions of O, S, Zn, Fe, Pb, and Si in the partially reacted sphalerite particle could provide some helpful Information about the oxidation mechanism of the zinc sulfide ores during roasting. It can be observed that during the roasting process, the sulfur in the sphalerite particle is gradually replaced by oxygen (Figure 3c), and the sulfur originally present in the reacted portion has disappeared (Figure 3d). The zinc remains in-situ and reacts with oxygen and iron to form a (Zn, Fe)O phase (Figure 3e). As the iron gradually migrates to the periphery of the particle (Figure 3f), the ZnFe_2_O_4_ phase is finally formed on the outer reacted portion [25]. Additionally, a trace amount of lead (Figure 3g) and silicon (Figure 3h) are detected in specific points of the particle. The lead is detected as Fe-Pb-Zn-Si-O phase enveloped by a thin shell of Fe-Zn-Si-O phase. Combined with the above results in Figure 2a (particle 4), the lead and silicium are more likely to be present in the original sphalerite particle instead of the oxide fume deposited on the particle. During roasting, the Zn_2_SiO_4_ is formed by the solid reaction between ZnO and silicates [30], and a trace amount of Pb-Zn silicate is formed in the portion where Pb is distributed. By the differences in the densities of the sphalerite and its reaction products and the evolution of SO_2_ gas during roasting, the reacted portion is gradually decomposed and turns into porous or tiny particles [25].

### 3.2. Estimate of the Roasting Temperature

To determine the original roasting temperature, a comprehensive thermal analysis (TG-DTA) was conducted on the unearthed zinc calcine (DZC-1). The thermal analysis curve (Figure 4) shows a noticeable weight loss between 256 and 357 °C accompanied by two endothermic peaks. In contrast, another distinct weight loss appears at temperatures higher than 847 °C. Since the unearthed calcines have been buried for an extended period after roasting, they may have been impacted by the ambient environment. One possible consequence is that phase transformation in the calcines took place during the long burial process.

Combined with the TG-DTA analysis results, the reheating temperature was selected as 200, 300, 500, 700, and 900 °C, respectively. As it can be seen in Figure 5, the phase composition of the zinc calcine begins to change after reheating at 300 °C. On one hand, hemimorphite (Zn_4_Si_2_O_7_(OH)_2_(H_2_O)) loses its water of crystallization, turning into dehydrated hemimorphite (Zn_4_Si_2_O_7_(OH)_2_). On the other hand, cerussite (PbCO_3_) is completely decomposed and combines with water molecules to form lead oxide hydrate (PbO·xH_2_O). After heating at 500 °C, PbO·xH_2_O loses water of crystallization to turn into PbO, and partially oxidizes to Pb_3_O_4_. After heating at 700 °C, the phase compositions of the zinc calcine change significantly. Among them, Zn_4_Si_2_O_7_(OH)_2_ loses structural water [31] and is transformed into Zn_2_SiO_4_. The PbO and Pb_3_O_4_ phases disappear. In addition, new phases like larsenite (PbZnSiO_4_) and lead oxysulfate (Pb_2_SO_5_) appear. However, the particle morphology of the reheated calcine is still consistent with that of the unearthed zinc calcine before heating, keeping a certain angular shaped morphology of the original particle. After heating at 900 °C, the PbSO_4_ grains in the zinc calcine completely disappear and transform into lanarkite (Pb_2_O(SO_4_)). In addition, fluorite (CaF_2_) in the zinc calcine partially reacts to form a large amount of lead fluoride silicate sulfate (Pb_10_(SiO_4_)_3_(SO_4_)3F_2_) and a small amount of calcium sulfate (CaSO_4_). At this temperature step, the microstructure of the zinc calcine has also undergone significant changes (Figure 6). The lead-containing particles appear to be homogenized. Moreover, the angular-shaped morphology of the calcine particles becomes crudely spherical with increased porosity.

Combined with the TG-DTA analysis result (Figure 4) and the XRD analysis results (Figure 5), the thermal weight loss between 256 and 357 °C should be mainly due to the crystal water loss of hemimorphite and the decomposition of cerussite. We would like to mention that the formation temperature of franklinite (ZnFe_2_O_4_) is approximately 600 °C [23]. Given the weathering effect during the burial process, it is reasonable to consider that both cerussite and hemimorphite could be weathering products [21] rather than phases formed after roasting. The main reason for the obvious weight loss at around 847 °C on the TG-DTA curve may be the decomposition of PbSO_4_ and the volatilization of PbO. Therefore, it can be inferred that the original roasting temperature of the unearthed calcines should be above the formation temperature of ZnFe_2_O_4_ and lower than the apparent weight loss temperature on the TG-DTA curve at about 850 °C.

### 3.3. Experimental Reconstruction of the Roasting Process

The WDXRF and XRD analysis results of the unearthed zinc calcine, the ore used for the simulation experiments, and the samples obtained under different simulated conditions are shown in Table 1. According to the analytical results, the ores used in the simulation experiments and the zinc calcine unearthed from the Doulingxia site contains a large amount of Fe, Zn, and Pb. The main mineralogy of the ore used for the simulation experiments includes sphalerite ((Zn, Fe)S), galena (PbS) and siderite (Fe(CO_3_)), as well as minor amounts of pyrrhotite (Fe_7_S_8_), pyrite (FeS_2_), and marcasite (FeS_2_). It is noted that the sulfur content in the unearthed calcine is quite low (0.33 wt.%), which indicates that the roasting process was carried out thoroughly. 

In order to verify the roasting temperatures of the unearthed zinc calcine estimated through the above reheating experiments, the simulated roasting temperature was further selected as 550, 650, 750, 850, and 950 °C, respectively. Firstly, the ores were roasted at 650 °C for 1, 6, 12, 24, 48, 96, and 192 h, respectively, to evaluate the dependence of the roasting time on the roasting process. As shown in Figure 7a, the degree of desulfurization increases significantly at 650 °C. After 24 h of roasting treatment, the sulfur content in the roasted products is reduced from 10.6 to 2.15 wt.% (Table 1). The experiments clearly show that the degree of desulfurization almost keeps stable after 24 h (Figure 7a). Combined with the XRD results, it is concluded that the following basic chemical reactions can be proposed.

Firstly, in addition to the decomposition of the associated siderite (FeCO_3_), the pyrite, marcasite, and pyrrhotite are oxidized as described in Equations (2)–(4) below.
(2)$$4{\mathrm{FeCO}}_{3}\left(s\right)+O_{2}\left(g\right)\to 2{\mathrm{Fe}}_{2}O_{3}\left(s\right)+4{\mathrm{CO}}_{2}\left(g\right)$$
(3)$$4{\mathrm{FeCO}}_{3}\left(s\right)+O_{2}\left(g\right)\to 2{\mathrm{Fe}}_{2}O_{3}\left(s\right)+4{\mathrm{CO}}_{2}\left(g\right)$$
$$2\mathrm{ZnS}\left(s\right)+3O_{2}\left(g\right)\to 2\mathrm{ZnO}\left(s\right)+2{\mathrm{SO}}_{2}\left(g\right)$$
(4)$$4{\mathrm{Fe}}_{7}S_{8}\left(s\right)+53O_{2}\left(g\right)\to 14{\mathrm{Fe}}_{2}O_{3}\left(s\right)+32{\mathrm{SO}}_{2}\left(g\right)$$

With the proceeding of the roasting process, galena and sphalerite are oxidized to lead sulfate and zinc oxide, respectively, as described in Equations (5) and (6) below.
(5)$$\mathrm{PbS}\left(s\right)+2O_{2}\left(g\right)\to {\mathrm{PbSO}}_{4}\left(s\right)$$
(6)$$2\mathrm{ZnS}\left(s\right)+3O_{2}\left(g\right)\to 2\mathrm{ZnO}\left(s\right)+2{\mathrm{SO}}_{2}\left(g\right)$$

We would like to mention that iron is usually present in sphalerite’s crystal lattice, substituting zinc. Therefore, if the oxidation reaction of the sphalerite takes place, sulfur in the sulfide can be replaced by oxygen to produce zinc oxide, which could react with coexisting Fe_2_O_3_ to produce franklinite through a solid-phase reaction [32,33], as described in Equation (7) below.
(7)$$\mathrm{ZnO}\left(s\right)+{\mathrm{Fe}}_{2}O_{3}\left(s\right)\to {\mathrm{ZnFe}}_{2}O_{4}\left(s\right)$$

However, it is noted in Table 1 that no ZnFe_2_O_4_ can be detected until the roasting time is 24 h. It is believed that the formation of franklinite is a solid-state reaction process between zinc oxide and iron oxide. Although the close contact of ZnO with Fe_2_O_3_ can facilitate the solid-phase reaction, the diffusion of the solid-state particles is still slow. In particular, the produced new intermediate phase can further efficiently hinder the atomic diffusion process [34,35]. It should be pointed out that there is a large amount of franklinite in the unearthed zinc calcine as well. From this view, it is believed that the unearthed zinc calcine should have been obtained by a lengthy roasting process with a duration not less than 12 h.

In the present simulation investigation, the maximum duration time for the roasting experiments was set at 48 h to ensure a fully completed roasting process. As expected, the results show that the degree of desulfurization gradually increases with the increase in the roasting temperature (Figure 7b). When the roasting temperature increases to 650 °C from 550 °C, the degree of desulfurization increases most significantly (Figure 7b). In contrast, when the roasting temperature is between 650 and 950 °C, the degree of desulfurization increases slowly. Considering the XRD results, at 550 °C, even after roasting for 48 h, sphalerite has not been completely oxidized. However, no zinc oxide phases such as ZnO and ZnFe_2_O_4_ can be detected, as seen in Table 1. After roasting at 950 °C for 48 h, the sulfur content in the roasted product can be as low as 0.38 wt.%. However, it is observed that the product turns black, which is obviously different from the unearthed zinc calcine. It is due to a large amount of magnetoplumbite (PbFe_12_O_19_) that is formed in the roasted product and cannot be detected in the unearthed calcine. Overall, when the roasting temperature is in a range of 650–850 °C, the roasted products by reconstruction experiments are similar to the zinc calcine unearthed at the Doulingxia site.

To make a better comparison with the unearthed zinc calcine, the products roasted for 48 h at 650, 750, and 850 °C were crushed and roasted for three cycles in the present reconstruction experiments. According to the results, the degree of desulfurization of the products almost remains constant when roasted again at 650 °C. In contrast, roasting for more cycles at 750 and 850 °C can further improve the degree of desulfurization of the products (Figure 7b). In particular, even after roasting for three cycles at 650 °C, a large amount of galena can still be detected by XRD analysis. As shown in Figure 8a, SEM images show that most of the final product obtained at 650 °C still retains the angular shaped morphology of the original particles. Furthermore, most of the PbSO_4_ particles show a layered porous structure, sometimes covering the unreacted galena (Figure 8a). In comparison, the mineralogical composition and microstructure of the final product at 750 °C, as revealed by SEM analyses, is very close to that of the unearthed zinc calcine. Zinc mainly occurs as franklinite and zincite, while lead exists in the form of lead sulfate (Table 1). The angular structure of the particles in the final product at 750 °C becomes less obvious, and several spherical porous particles appear in the image (Figure 8b). By contrast, the final roasting product at 850 °C contains a small amount of lanarkite (Pb_2_OSO_4_), which has not been detected in the unearthed zinc calcine. Moreover, the microstructure of the final product obtained at 850 °C is dominated by spherical porous particles with relatively large pores (Figure 8c). Therefore, it is reasonable to deduce that the roasting temperature of the unearthed zinc calcine should be between 650 and 850 °C, most probably near 750 °C.

The final roasting products obtained at 750 °C are mainly composed of a major amount of hematite (Fe_2_O_3_), franklinite (ZnFe_2_O_4_), zincite (ZnO), and anglesite (PbSO_4_), as shown in Table 1. By contrast, major amounts of willemite (Zn_2_SiO_4_) and cerussite (PbCO_3_), as well as minor amounts of hemimorphite [Zn_4_Si_2_O_7_(OH)_2_(H_2_O)] and fluorite (CaF_2_) can be found in the unearthed zinc calcine. The presence of Zn_2_SiO_4_ in the unearthed calcines mainly contributed by the higher content of Si (Table 1). Fluorite should come from gangue because of its chemical stability. Combined with the reheating experimental results, it confirms that PbCO_3_ and Zn_4_Si_2_O_7_(OH)_2_(H_2_O) found in the unearthed zinc calcine should have been formed by weathering.

It is noted that in the unearthed zinc calcine, zinc mainly exists in the form of ZnFe_2_O_4_. The reduction in zinc from the ZnFe_2_O_4_ is much more complicated than that from the ZnO. In a reducing atmosphere, ZnFe_2_O_4_ can be reductively decomposed into ZnO and Fe_2_O_3_ at about 700 °C [36]. As can be inferred from the Ellingham Diagram [24], iron oxides were reduced more easily than zinc oxide. A large amount of iron oxide in the calcine will increase the consumption of reducing coal. In addition, the metal iron that appears in the slag could lower its gas permeability. The metal iron can even adhere to the inner wall of the retorts, affecting the slag tapping process and the reutilization of the retorts. According to the analysis results of the slags at the Tongmuling site, slags tipped out from the pots present as loose granular particles, and only a small amount of iron can be found in the slags attached inside the pots [24]. It is likely that sufficient reducing coal had been added in the pots to prevent the formation of molten slag during the distillation processing [19,28]. Therefore, the presence of ZnFe_2_O_4_ does not bring too much negative influence on the entire reduction process and the final yield of zinc.

## 4. Conclusions

Combining the microstructure and mineralogical characterization of the unearthed calcine with the experimental reconstruction, the roasting technology in pre-modern zinc metallurgy production from zinc sulfide ores has been clarified and reconstructed. The original roasting temperature of the unearthed zinc calcine should be in a range of 650~850 °C, most probably near 750 °C. If the retention time is long enough, all sphalerite can be oxidized when the roasting temperature is above 650 °C. In the final roasted products, zinc mainly exists in the form of ZnFe_2_O_4_, ZnO, and Zn_2_SiO_4_ phases, while the lead exists in the form of PbSO_4_. PbCO_3_ and Zn_4_Si_2_O_7_(OH)_2_(H_2_O) found in the unearthed zinc calcine are considered as the weathering product during its burial process. The final roasting product is mainly tiny porous particles, which is beneficial for further distillation. These findings are helpful to completely reconstruct the ancient zinc metallurgy technology of zinc sulfide ores.

## Figures and Tables

**Figure 1 materials-14-02087-f001:**
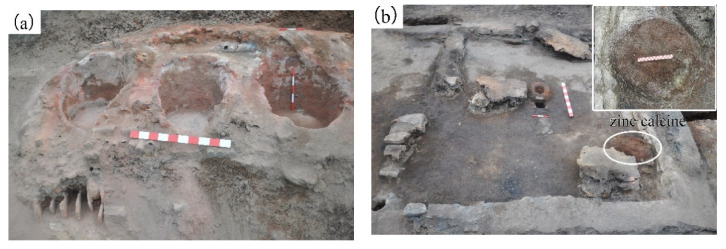
Archaeological evidence found in the Doulingxia site. (**a**) Cylindrical roasting hearths; (**b**) the unearthed zinc calcine.

**Figure 2 materials-14-02087-f002:**
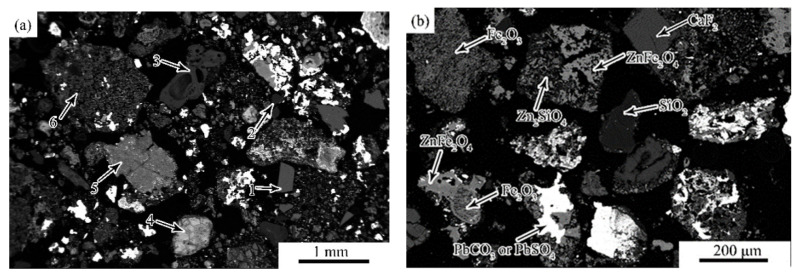
Backscattered electron images showing the morphology of the zinc calcine found at the Doulingxia site. (**a**) Low magnification, (**b**) tiny particles.

**Figure 3 materials-14-02087-f003:**
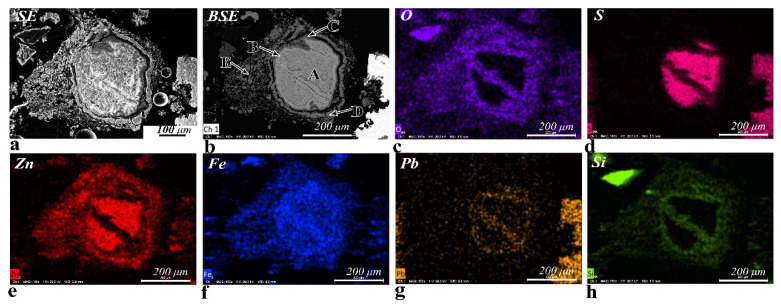
X-ray maps showing the distributions of O, S, Zn, Fe, Pb, and Si in a partially reacted sphalerite particle. (**a**) The secondary electron (SE) image shows the morphology of the grain; (**b**) Different grey levels in the backscattered electron (BSE) image imply different compositions. A—lightly reacted sphalerite (core), B—Fe-Pb-Zn-Si-O phase (brighter rim), C—Fe-Zn-Si-O phase (dark grey rim), D—a mixture of ZnFe_2_O_4_ and Zn_2_SiO_4_ (grey rim), E—a mixture of ZnFe_2_O_4_ and Zn_2_SiO_4_ (porous portion); (**c**) O Kα map; (**d**) S Kα map; (**e**) Zn Kα map; (**f**) Fe Kα map; (**g**) Pb Lα map; and (**h**) Si Kα map.

**Figure 4 materials-14-02087-f004:**
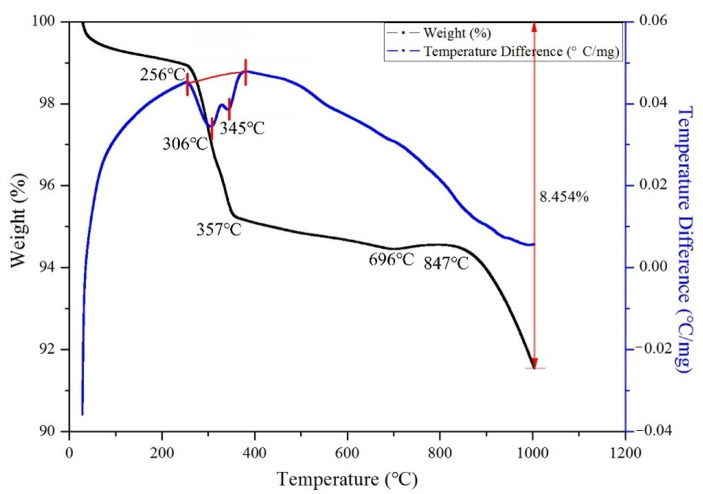
TG-DTA analysis result of the zinc calcine (DZC-1).

**Figure 5 materials-14-02087-f005:**
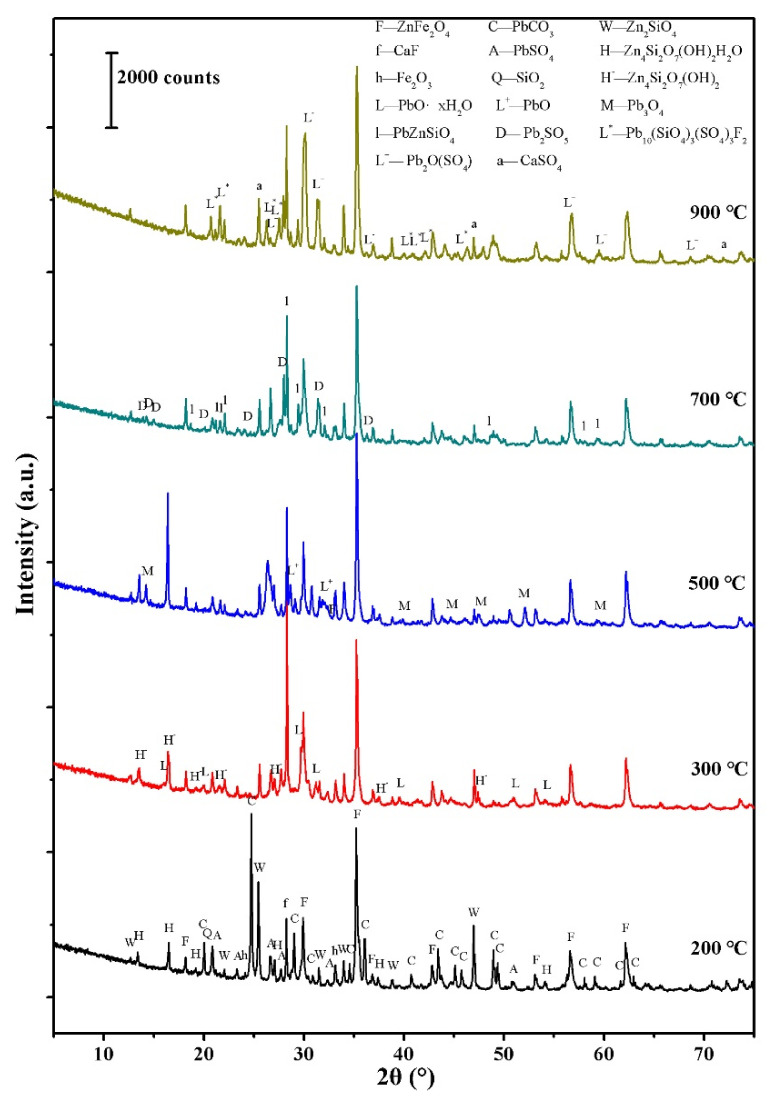
XRD analysis results showing the phase transformation of the zinc calcine (DZC-1) reheated at different temperatures.

**Figure 6 materials-14-02087-f006:**
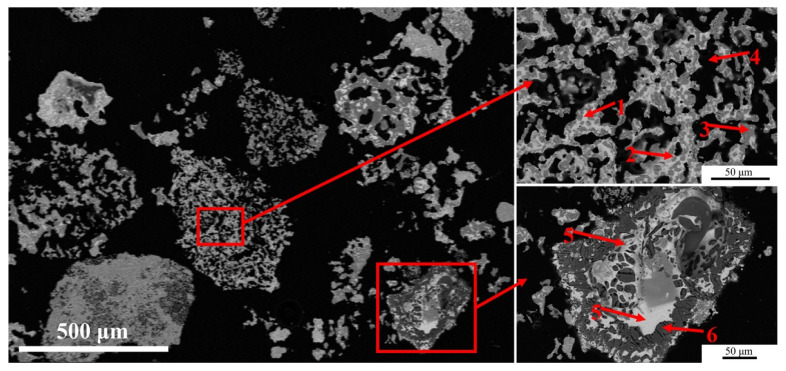
Backscattered electron image showing the morphology of the zinc calcine reheated at 900 °C. 1—PbZnSiO_4_ (light), 2—ZnFe_2_O_4_ (grey), 3—Zn_2_SiO_4_ (dark grey), 4—pore (black), 5—Pb_10_(SiO_4_)_3_(SO_4_)_3_F_2_ (light), 6—CaSO_4_ (dark).

**Figure 7 materials-14-02087-f007:**
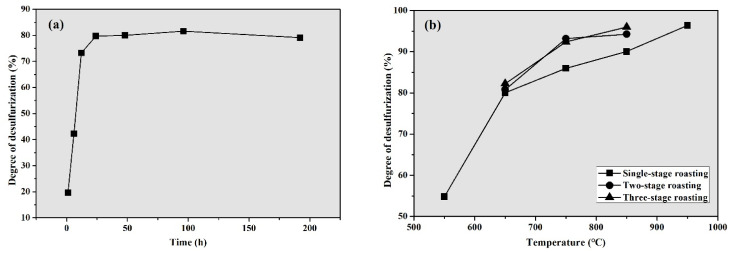
Degree of desulfurization of the products obtained under different conditions. (**a**) Products roasted at 650 °C for different roasting times, (**b**) products roasted for 48 h at different temperatures.

**Figure 8 materials-14-02087-f008:**
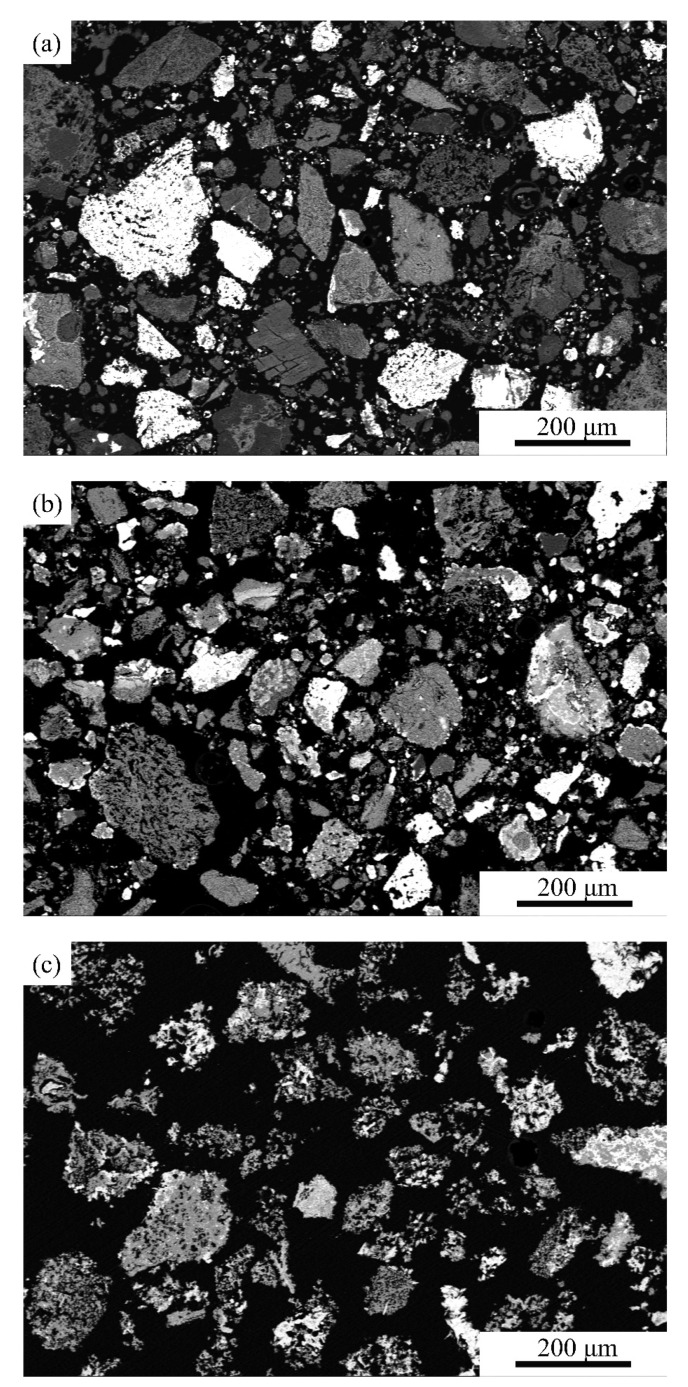
Backscattered electron images showing the morphology of the final products obtained under different simulated (**a**) 650 °C; (**b**) 750 °C; (**c**) 850 °C.

**Table 1 materials-14-02087-t001:** The WDXRF and XRD analysis results of the unearthed zinc calcine, the ore used for the experiments, and the samples obtained under different experimental conditions.

No.	Chemical Contents (wt.%)						Phases Composition	
	O	S	Si	Fe	Zn	Pb	Major Phases	Minor Phases
DZC-1	14.48	0.33	2.19	29.01	26.97	16.90	franklinite (ZnFe_2_O_4_), cerussite (PbCO_3_), willemite (Zn_2_SiO_4_)	hemimorphite(Zn_4_Si_2_O_7_(OH)_2_(H_2_O), anglesite (PbSO_4_), fluorite (CaF_2_), quartz (SiO_2_), hematite (Fe_2_O_3_)
Hsp-1	8.42	10.64	1.12	26.85	24.67	23.40	sphalerite ((Zn,Fe)S), galena (PbS), siderite (Fe(CO_3_))	pyrrhotite (Fe_7_S_8_), pyrite (FeS_2_), marcasite (FeS_2_)
650-1 h	12.53	8.55	1.30	39.76	17.45	13.57	sphalerite, galena	pyrrhotite, hematite, orthoclase (KAlSi_3_O_8_)
650-6 h	13.66	6.14	1.22	38.08	18.63	15.58	sphalerite, galena, hematite	anglesite, pyrrhotite, sanidine (KAlSi_3_O_8_)
650-12 h	14.85	2.84	1.24	35.50	19.00	20.90	sphalerite, galena, hematite	anglesite, pyrrhotite
650-24 h	15.64	2.15	0.84	38.39	13.71	23.46	sphalerite, galena, hematite, anglesite	zincite (ZnO), franklinite
650-48 h	15.40	2.13	0.83	39.43	13.83	22.58	sphalerite, galena, hematite, anglesite, franklinite	zincite, fluorite
650-96 h	16.53	1.95	0.93	37.16	15.79	21.99	anglesite, hematite, franklinite, galena, zincite	zinc oxide sulfate (Zn_3_O(SO_4_))
650-192 h	15.97	2.22	1.33	35.93	13.93	24.61	anglesite, hematite, franklinite, galena	zincite, zinc oxide sulfate
550-48 h	14.51	4.81	0.61	37.67	18.04	19.34	sphalerite, galena, hematite	anglesite
750-48 h	15.29	1.49	0.48	28.03	23.54	26.83	anglesite, hematite, franklinite, zincite, galena	
850-48 h	13.06	1.06	0.50	32.29	24.95	24.06	franklinite, zincite, hematite, anglesite, lanarkite (Pb_2_OSO_4_)	
950-48 h	12.22	0.38	0.41	38.21	27.41	16.55	franklinite, zincite, magnetoplumbite (PbFe_12_O_19_)	hematite, willemite
650-48 h-2	15.21	2.04	0.61	36.15	12.80	27.44	anglesite, hematite, franklinite	zincite, galena, zinc oxide sulfate
750-48 h-2	14.92	0.73	1.75	42.49	19.91	13.79	hematite, franklinite, zincite, anglesite	orthoclase
850-48 h-2	13.45	0.61	1.02	37.51	22.89	17.38	franklinite, zincite, hematite, lanarkite	willemite, anglesite
650-48 h-3	15.17	1.89	0.77	37.79	16.63	22.36	anglesite, hematite, franklinite, galena, zincite	zinc oxide sulfate
750-48 h-3	14.81	0.81	0.99	38.67	21.37	17.93	hematite, franklinite, zincite, anglesite	orthoclase
850-48 h-3	13.84	0.43	0.58	39.63	22.21	17.06	franklinite, zincite, hematite	lanarkite, willemite

The WDXRF results were normalized to 100 wt.%. DZC-1 refers to the zinc calcine unearthed in the Doulingxia site. Hsp-1 refers to the ore used for reconstruction experiments. 650-1 h means roasting at 650 °C for 1 h. 650-48 h-2 means roasting at 650 °C for 48 h twice. The numbering methods of the other samples can be similarly deduced.

## Data Availability

Data is contained within the article.

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
