# Peer review of "Microstructure, Mineralogical Characterization and the Metallurgical Process Reconstruction of the Zinc Calcine Relics from the Zinc Smelting Site (Qing Dynasty)"

_materials, 2021, doi:10.3390/ma14082087_

Round 1
Reviewer 1 Report
Very good paper providing information on the conditions used to roast zinc ores in historical smelting processes. Only minor suggestions.
On p.6, line 7 it says "oxygen atoms diffuse out of the sphalerite". Surely it should say "sulphur dioxide", not oxygen?
Please ensure mineral names in the text do not start with upper case letters, e.g. Pyrrhotite on p. 9
Reviewer 2 Report
Please find my comments and corrections in the enclosed file

Reviewer 3 Report
Ya Xia et al present a particularly interesting and innovative article in terms of the history of techniques. Indeed, it is one of the rare studies on the conditions of transformation of sphalerite in the context of an old foundry. They propose a very good approach to understand the phenomena in play. They also provide information on the lead production chain from galena. The conclusions remain conservative and are well supported by the data provided.
The remarks are few:
In the abstract, can you clarify the dating of the site? As it stands, it is understood that it is used from 1636 to 1912.
In the abstract, the temperature data is provided but not the duration. This data should be added.
Page 2: "The archaeological evidence and scientific analysis showed that the ores distilled in Chongqing, southwest China, in the Ming CE 1368-1644) and Qing (CE 1636-1912) dynasties, were mainly oxidized ores." Is there only one archaeological evidence?
At the level of Figure 1, an archaeological plan of the furnaces from the Doulingxia site should be added
Page 6 : " lead and silicon " : Maybe we should change silicon by silicium to avoid confusion.
On figure 6 (p.9), the numbers of the different crystal structures on the image should be made more readable.
Page 14, The use of the term "slag" is abusive. The process in question is similar to a mineralurgical treatment (ore dressing process). No slag is produced unless intentional smelting takes place. It would be better to use the term "mineral waste".
Round 2
Reviewer 2 Report
The revised version is much clearer and scientifically correct. Apart from a few, very minor correction (see the enclosed draft), the paper can be accepted for publication in Materials.
